# Large Extracellular Vesicles—A New Frontier of Liquid Biopsy in Oncology

**DOI:** 10.3390/ijms21186543

**Published:** 2020-09-07

**Authors:** Gaetano Pezzicoli, Marco Tucci, Domenica Lovero, Franco Silvestris, Camillo Porta, Francesco Mannavola

**Affiliations:** 1Department of Biomedical Sciences and Human Oncology, University of Bari ‘Aldo Moro’, 70121 Bari, Italy; pezzicoligaet@gmail.com (G.P.); marco.tucci@uniba.it (M.T.); dom.lovero@gmail.com (D.L.); francesco.silvestris@uniba.it (F.S.); camillo.porta@uniba.it (C.P.); 2National Cancer Center, Tumori Institute Giovanni Paolo II, 70121 Bari, Italy

**Keywords:** large extracellular vesicles, microvesicles, oncosomes, exosomes, cancer

## Abstract

Extracellular Vesicles (EVs) are emerging as pivotal elements in cancer. Many studies have focused on the role of Small- (S)-EVs but in recent years Large-(L)-EVs have progressively gained increasing interest due to their peculiar content and functions. Tumor-derived L-EVs carry a lot of oncogenic proteins, nucleic acids and lipids to recipient cells and are involved in the reshaping of the tumor microenvironment as well as in the metabolic rewiring and the promotion of the pro-metastatic attitude of cancer cells. Several techniques have been developed for the isolation of L-EVs and commercial kits are also available for efficient and easy recovery of these vesicles. Also, the improvement in DNA sequencing and “omics sciences” profoundly changed the way to analyze and explore the molecular content of L-EVs, thus providing novel and potentially useful cancer biomarkers. Herein, we review the most recent findings concerning the role of L-EVs in cancer and discuss their possible use in oncology as “liquid biopsy” tools as compared to the other classes of EVs.

## 1. Introduction

Extracellular Vesicles (EVs) are emerging as important intercellular messengers regulating tumor progression and cell metabolism. Based on diameter, different classes of EVs are known, including either Small- (S) or Large- (L)-EVs. Among the S-EVs, exosomes are nano-sized vesicles deriving from the late endocytic pathway, carrying a large variety of molecules, such as proteins, DNA fragments and RNAs [1]. In the past few years, exosomes have been intensively investigated in cancer models to unmask new therapeutic targets or putative circulating biomarkers [2,3,4]. Despite all the discoveries made, few have been translated into the clinic, mainly due to the lack of standardized methods for either nanovesicle isolation or downstream analyses. Moreover, suitable yields of S-EVs for downstream analyses still require large volumes of biological fluids, thus limiting the applicability of exosomes as a high-throughput diagnostic tool due to high costs and analytical time.

Although their role is not completely understood, L-EVs have progressively gained increasing interest in the last years. Recently, L-EVs have been proposed to be involved in cancer metabolic rewiring [5], as well as in chemotherapy resistance [6]. Their particular composition and easiness of recovery from any biological fluid make them a promising alternative to S-EVs, particularly as a circulating diagnostic tool for molecular analyses.

Here, we summarize the main characteristics of EVs with a particular focus on L-EVs and discuss their involvement in cancer progression, as well as their possible applications in oncology. 

## 2. Nomenclature of Extracellular Vesicles

In 2018 the International Society for Extracellular Vesicles revised the definition of EVs as “particles naturally released from the cells that are delimited by a lipid bilayer and cannot replicate” [7]. However, there is not univocal consensus about the nomenclature of different EV subtypes and the lack of specific markers makes their exact differentiation quite difficult. Therefore, as a result of the “Minimal information for studies of extracellular vesicles 2018” (MISEV2018) position statement, EVs should be named based on their size, biochemical composition and cell of origin. Those that are less than 200 nm in diameter are commonly referred to S-EVs, or exosomes, while EVs that exceed this size are called medium- (M) or L-EVs and include microvesicles, shedding bodies, ectosomes and microparticles. Further classification is based on the biochemical composition depending on the expression of either surface or intracellular markers, including tetraspanins, Annexin-V or heat-shock proteins (e.g., CD63/CD81^+/−^, ANXV^+/−^ or HSP70^+/−^ vesicles). Another possible nomenclature is based on the EV origin, such as podocyte- or platelet-derived EVs, as well as oncosomes or apoptosomes, for L-EVs released from cancer or apoptotic cells, respectively.

This nomenclature was structured to organize the chaotic literature on EVs. However, the EV name was misused or improperly used in many papers, making it hard to correctly define EV classes due to the lack of proof of their origin.

## 3. Biogenesis of L-EVs

Despite the biology of S-EVs has been widely explored [1], L-EVs’ biogenesis is poorly understood and their relevance has been only recently acknowledged.

They originate from the outward budding of the plasma membrane, which is guided by specific molecular machinery [8]. One of the major mechanisms includes the interaction between the membrane Arrestin Domain-Containing Protein-1 (ARRDC1) with the TSG101 late endosomal protein, leading to the relocalization of TSG101 from endosomal compartment to the plasma membrane (Figure 1). This event provokes a shape change in the cell membrane curvature with consequent release of microvesicles outside the extracellular space in a fashion similar to that used by viruses to bud off the cell [9]. Specific classes of lipids also play a role in L-EVs’ budding. The formation of ceramide by acid sphingomyelinase and the translocation of phosphatidylserine on the outer membrane layer are early events for the generation of microvesicles and plasticity of their membrane [10,11,12].

It has been debated whether L-EVs are waste cellular products or if they exert specific functions in relation to their content. The second hypothesis is more likely to be correct, since a number of proteins involved in this active loading have been identified. The most widely studied is the ADP-ribosylation factor 6 (ARF6), namely a protein expressed on the surface of endosomes (Figure 1). Although its role is unknown, ARF6 may influence the incorporation of integrins and the major histocompatibility complex (MHC)-I into microvesicles. Moreover, this protein is involved in recruiting the myosin light-chain kinase (MLCK) via ERK, thus favoring the interaction between actin and myosin that initiate the outward budding of the plasma membrane [13].

The biogenesis of L-EVs in cancer cells follows the same mechanisms described for normal cells, although some differences were reported for large oncosomes. Notably, the formation of oncosomes is prompted by the disruption of both cytoskeletal and membrane plasticity regulators as recently demonstrated by Minciacchi et al. [5]. Diaphanous-related formin-3 (DIAPH3), for example, is a cytoskeletal regulating protein whose inactivation triggers the generation of membrane blebs of 1–10 μm in diameter (Figure 1). Several gene expression modifications of the key regulators of either cell proliferation, apoptosis and migration, have been linked to DIAPH3 deregulation, such as those affecting the Caveolin-1, MyrAkt1 and the soluble form of the Heparin-binding (HB)-EGF-like growth factor [14,15,16].

## 4. Content of L-EVs

### 4.1. Nucleic Acids

One of the most studied topics of L-EVs is the exploration of their nucleic acid content, which is pretty unique concerning all the other EVs. With respect to S-EVs, oncosomes contain up to 7-fold more DNA and structural analyses showed that it is condensed in a chromatinized form, with large DNA fragments ranging from 100 Kbps to 2 Mbps [17]. Interestingly, a high concordance between the mutational status of L-EV-DNA and that of parental cells was evidenced. The L-EV-DNA covered the entire reference genome of their donor cells spanning all chromosomes, thus suggesting that the L-EV genome may be a bona fide representation of the cancer genome. Tumor-specific somatic copy-number variation was documented in L-EVs isolated from plasma of mice models with PCa bone metastasis, thus opening to the possibility of performing liquid biopsy with L-EVs isolated from blood [17].

Since tumor-derived RNAs packaged within the phospholipidic bilayer of L-EVs are protected from degradation by ribonucleases [18], their analyses may also provide additional information on originating cells with either potential diagnostic or prognostic purposes. While lacking a specific mRNA content, L-EVs have been reported to be enriched of several classes of small RNAs, such as miRNAs, transfer (t)-RNAs, ribosomal (r)-RNAs, vault (v)-RNAs, small nuclear (sn)-RNAs and small nucleolar (sno)-RNAs. Hong et al. found that the content of RNA molecules in colorectal cancer-derived L-EVs was increased with respect to that from S-EVs. The greater part of L-EV-RNA was composed of miRNAs and other short regulatory RNAs, while coding-RNAs were less present. Moreover, they identified specific miRNA signatures for each class of EVs analyzed, suggesting the use of L-EV miRNAs as biomarkers with diagnostic purposes [19].

### 4.2. Proteins 

Differences in protein content have been reported between tumor-derived S-EVs and L-EVs with some classes of proteins found up to 60-fold increase in L-EVs. In particular, enzymes implied in glucose (GAPDH), glutamine (GLS) and aspartate (GOT1) metabolism were found constantly up-regulated in L-EVs, suggesting their involvement in the metabolic rewiring of recipient cancer cells [5]. Additionally, tumor-derived L-EVs are typically engulfed with proteins implied in the Calnexin-Calreticulin cycle, driving post-translational gene expression regulation.

To a minor extent, L-EVs also carry proteins implied in DNA packaging (Histone H2B type 2F), angiogenesis (Glycoprotein I), apoptosis control (Voltage-dependent anion channels 2) and cell migration (mitochondrial ATP synthase F1 subunit β) [5]. Cytokeratin (CK)-18 is considered a typical L-EV protein since it is particularly enriched only in this class of EVs [5]. As CK-18 is frequently expressed in the cell membrane proximity [20], it is possible that L-EVs passively incorporate this protein during the generation of the plasmatic blebs. However, an active loading of this structural protein for stabilizing both form and shape of L-EVs is also not excluded, thus conferring stress-resistance.

### 4.3. Lipids

Several fatty acids are enriched in L-EVs, either with a structural or functional role, including the ceramide and phosphatidylserine implied in the biogenesis of all kinds of vesicles [10,11,12,21]. Active signaling lipids are also present in the phospholipid membrane of L-EVs. For example, ATP-stimulated microglia can release L-EVs containing the endocannabinoid molecule N-arachidonyl-ethanolamine (anandamide), which activates the Cannabinoid Receptor-1 (CB1) and induce metabolic changes in the recipient cells [22].

It was recently demonstrated that cells can modulate the lipid expression on L-EVs, depending on the function they are committed to. For example, platelet-derived L-EVs at the inflammatory sites can convert their membrane lipids into the signal molecule 12-HETE (12-hydroxy eicosatetraenoic acid) through the extracellular phospholipase A2 and the vesicular 12-lipoxygenase [23]. The 12-HETE seems to play a role in the L-EVs’ internalization by neutrophils, inducing their activation in situ. A similar mechanism could be used by cancer cells, since it has been demonstrated that a 12-HETE receptor, namely the BLT2, is implicated in cancer cell transformation, invasion and metastasis [24,25], as well as VEGF-mediated angiogenesis [26].

## 5. Methods for the Isolation of L-EVs

EVs can be isolated from either culture media and biological fluids, including blood, urine, saliva and ascites. The EV isolation techniques can be grouped into five distinct categories—ultracentrifugation, density-gradient separation, polymer-based precipitation, immune selection and microfluidic isolation [27] (Table 1). 

### 5.1. Ultracentrifugation

The classical method for isolating EVs is based on the separation of particles by centrifugation. Initially, the high buoyant density particles (e.g., cells, cell debris, apoptotic bodies and biopolymers aggregates) are pelleted at low-speed centrifugation. The non-EV proteins are then removed by resuspension of the pellet, followed by repeated high-speed centrifugation [28]. The efficiency of EV isolation by centrifugation depends on many factors, such as viscosity of the sample, acceleration (*g*) and type of rotor [29]. While the isolation of S-EVs requires high acceleration (100,000× *g*), L-EVs isolation is performed at lower acceleration. Many authors reported that consistent isolation can be performed at 10,000× *g* and this makes the isolation of L-EVs simpler [5].

Although centrifugation is time-consuming, it needs a relatively small set of reagents and consumables, as well as it can be easily applied to large starting volumes. Although it produces consistent yields of L-EVs, they are generally tainted by a lot of contaminating proteins, such as albumin for plasma isolated EVs or uromodulin/Tamm–Horsfall protein (THP) for vesicles of urinary derivation. Thus, adequate controls are needed to verify the purity of harvested vesicles, since inadequate samples may affect downstream analyses of functional experiments [7].

### 5.2. Ultrafiltration

It is based on the use of commercial membrane filters with pores of various sizes. The EV preparation goes through the membrane, losing all the components larger than the pores. Large particles are removed first by filters with broad pore diameters and the particles with a size smaller than the target EVs are separated from the filtrate at the next stage. Further ultracentrifugation stages can be added and the yields are inversely proportional to the number of filtration stages, while directly proportional to the purity of EV preparations [30,31]. A great limit of this method is the high pressure needed during the filtration steps, which may lead to a leakage of contaminant particles or rupture of EVs [32].

### 5.3. Gel Filtration (Size Exclusion Chromatography)

Gel filtration is based on the separation of molecules according to their hydrodynamic radius and it is largely used for separating biopolymers (e.g., proteins, polysaccharides, proteoglycans, etc.). This method is also applicable to the separation of EVs from both plasma and urine protein complexes as well as lipoproteins [33]. Briefly, the EV solution is passed through a gel column. The gel has a peculiar structure that causes some analytes to reach the base of the column faster than others. In this way, EVs are divided from other components and may be separately eluted. For a simple EV isolation by gel chromatography, several types of commercial columns are available: qEV Size Exclusion Columns (Izon Science Ltd., Oxford, UK), Sepharose 2B (Sigma, St. Louis, MO, USA), Sepharose CL-4B (Sigma, St. Louis, MO, USA). This latter method is very user-friendly but expensive. Moreover, despite the high purity of EV preparations obtainable with gel chromatography, the final concentration of particles is mostly diluted and can be very low, thus requiring additional concentration steps prior to downstream applications [29]. Finally, it is worth noting that at the present moment, no commercial kit can offer the separation of different EVs classes, thus making this method not suitable for the precise isolation of L-EVs.

### 5.4. Precipitation

The second most popular method for EV isolation is based on precipitation technique through hydrophilic polymers. EVs are dispersed in a solution of super hydrophilic polymers that restrains their solubility. Thus, the vesicles can be sedimented by low-speed centrifugation (1500× *g*). Different polymers may be used such as polyethylene glycol, sodium acetate (which can neutralize negative charges on the EV membrane), protamine (a positively charged molecule) or, as recently proposed, the PRotein Organic Solvent Precipitation (PROSPR) [34]. Although this method is much faster than ultracentrifugation, it often implies the coprecipitation of non-EV nucleoproteins and proteins, such as albumin and apolipoprotein E [35]. It is generally used when a high amount of EVs is required, with no need for excessive purity. Similarly to gel filtration, this method does not offer the possibility to separate different EV classes and thus possible contamination by S-EVs should be considered when analyzing L-EVs isolated with precipitation.

### 5.5. Immune Affinity Interaction

The use of specific mAb, directed against typical EV surface molecules, such as tetraspanins, can be used in conjunction with magnetic beads, to selectively bind EVs [36]. Other possible tools used with similar purposes are highly porous monolithic silica microtips, plastic plates, as well as cellulose and membrane affinity filters. Some authors reported the efficient isolation of EVs with the use of magnetic beads coated with a pool of anti-CD9, anti-CD63, anti-CD81 and anti-EpCAM antibodies. Once captured, EV-magnetic bead complexes are then separated and washed by using a magnet [37]. This method offers the possibility to reduce the isolation time, elevate the purity of EV preparations and harvest specific EV fractions, including L-EVs [38]. On the other hand, this method is expensive and characterized by a low isolation efficiency, while processing large volumes is challenging [29].

### 5.6. EV Markers for Quality Control

Following isolation of L-EVs from either cell culture supernatants or biological fluids, quality controls are mandatory in order to schedule downstream analysis of EVs. For this purpose, typical assays are based on flow cytometry and Western blotting for protein content characterization, Tunable Resistive Pulse Sensing (TRPS) for dimensional assessment and transmission electron microscopy (TEM) for morphological study.

The MISEV2018 defined the criteria that must be fulfilled to demonstrate the EV nature and purity [7]. Specifically, the presence of at least (i) a transmembrane protein associated to the plasma membrane and/or endosomes (e.g., tetraspanins, MHC-class I, integrins, etc.) and (ii) a cytosolic protein typically recovered in EVs (ALIX, ESCRT, Flotillins and Caveolins), is required. Other assays are necessary when claiming specific classes of EVs. In the case of L-EVs, for example, several proteins associated with non-endosomal compartments (e.g., histones, cytochrome C or calnexin) need to be verified, since they are not present in S-EVs.

Finally, proteomics of the recovered EVs should be addressed to verify the parental origin of these vesicles or document their functional activity [7]. The presence of non-EV co-isolated contaminants, such as lipoproteins (APO1, fibronectin, collagen) for plasma and uromodulin for urine, must be also excluded to confirm the purity of EV preparations [7].

## 6. Role of L-EVs in Cancer

Tumor cells shed a high number of L-EVs as compared to normal cells [39]. Thus, it has been suggested that they may be implied in several processes regulating cancer cell proliferation and metastasis. Early studies investigating the role of L-EVs in cancer were completed in glioblastoma multiforme (GBM). Indeed, Al-Nedawi et al. showed that a truncated and oncogenic form of the EGFR, namely the EGFR variant III (EGFRvIII), can be exchanged between GBM cells through an intercellular transfer of membrane-derived microvesicles, with consequent accumulation on the cell surface [40]. This event leads to aberrant activation of the MAPK signaling pathway, with the consequent deregulation of EGFRvIII downstream genes, including the vascular endothelial growth factor (VEGF) and Bcl-xL anti-apoptotic up-regulation or the p27 cyclin-dependent kinase inhibitor down-regulation. Collectively, these changes drive the morphological transformations of GBM cells and confer increased anchorage-independent growth and pro-angiogenic capacity by reducing the apoptotic stimuli in recipient cells [40,41]. Moreover, L-EVs can transfer large DNA molecules to recipient cells to increase their tumorigenic potential, as demonstrated in vitro using HRAS-transformed rat epithelial cells. Vesicles released by these cells were found engulfed with chromatin-associated double-stranded DNA sequences, including the mutated form of HRAS [42]. The exposure to these L-EVs was able to mediate a stable transfection of the mutated form of HRAS into recipient wild type cells, with a consequent increase in the proliferative ability and phenotype changes.

Another described role of L-EVs includes angiogenesis and the reorganization of the tumor microenvironment. To this regard, other molecules carried by L-EVs include the vascular endothelial growth factor (VEGF), the tumor growth factor (TGF)-β and the microRNA (miRNA or miR)-1246, which are factors with known angiogenic potential. Noteworthy, the miR-1246 regulates the TGF-β/SMAD signaling pathway, promoting endothelial cell proliferation and vessel sprouting [43,44]. Bertolini et al. recently showed that GBM organoids can influence their surrounding microenvironment, modifying it toward a pro-tumorigenic state via EV-mediated transfer of the Vacuolar-ATPase subunit G1 and homeobox transcription factors. The authors hypothesized that these cargos were transferred through the L-EVs to both neoplastic and non-neoplastic glial recipient cells, thus mediating their reprogramming toward an oncogenic state as well as activating pathways driving proliferation and motility [45]. Similar findings were reported in an in vitro prostate cancer (PCa) model [46]. In this context, L-EVs were found to be internalized by different cell populations, including normal prostate fibroblasts. Stromal cells receiving PCa cell-derived L-EVs, increase their content of α-SMA, interleukin (IL)-6 and matrix metallopeptidase (MMP)-9, which were associated with the establishment of a pro-tumorigenic milieu. The release of proteolytically active MMP-containing L-EVs is thought to support the amoeboid-type invasion of tumor cells [47]. Moreover, L-EV-reprogrammed prostate fibroblasts were found to stimulate endothelial tube formation in vitro, while promoting tumor growth in mice. Comprehensively, these studies defined the contribution of L-EVs to the metastatic process, by preparing cancer cells for the detachment and dissemination from the primary tumor bed. To date, however, there is poor knowledge about the possible activities of L-EVs within distant sites from primary tumors, although a study demonstrated the presence of these vesicles in the peripheral blood of PCa-bearing mice [15]. Hence, it would be not surprising that circulating tumor-derived L-EVs also participate in the pre-metastatic niche formation, as already described for S-EVs [48]; further researches are required to further address this hypothesis.

The role of L-EVs in the interplay between tumor cells and the surrounding microenvironment is also another topic of interest, although at present is still poorly investigated. The bidirectionality of the EVs transfer between tumor cells and tumor stroma, in fact, has been recently demonstrated in gastrointestinal tumors since vesicles shed by cancer-associated fibroblasts (CAF) were found able to induce chemoresistance in either pancreatic [49] or colorectal cancer models [50]. Similarly, intratumoral mesenchymal stem cells (MSC) can release EVs that induce resistance to 5-fluorouracil-related apoptosis in gastric cancer cells [51]. These data, however, mainly refer to S-EVs, while a definite role of L-EVs needs to be verified.

Finally, the L-EVs also play a role in cancer immune escape. As recently reported, breast cancer (BC)-derived L-EVs can contain large amounts of indoleamine-2,3-dioxygenase (IDO) [52], an enzyme that is involved in tryptophan metabolism and plays a pivotal role in the establishment of an immune-suppressed microenvironment by dampening the T-cell proliferation [53]. Thus, it is likely that cancer cells exploit the L-EV-mediated release of IDO to wide-spread their suppressive signals within the tumor milieu, although other mechanisms may be also involved. On the other hand, Nanou et al. revealed that leukocytes are also able to release L-EVs [54]. Although the authors did not prove any evidence on their possible function, other studies demonstrated that leucocyte-derived EVs play an important role in the regulation of the leukocyte-endothelial interaction [55], thus suggesting a possible implication in tumor progression by favoring cancer cell intravasation and dissemination to metastatic sites.

## 7. L-EVs and Drug Resistance

The release of L-EVs is thought to function as a mechanism of acquired drug resistance exploited by cancer cells. In this regard, L-EVs can transfer the P-glycoprotein, also known as multidrug resistance protein (MDR)-1 and the multidrug resistance-associated protein (MRP)-1 from drug-resistant to drug-sensitive cancer cells [6]. The consequent overexpression of these two multidrug efflux transporters on the cellular plasma membrane works by efficiently clearing drugs off the intracellular compartment and is capable of conferring resistance to many chemotherapeutic drugs, in prostate [56,57], breast [58,59,60], lung [61] and ovarian [62] cancer, as well as in hepatocellular carcinoma [63]. Similarly, Ma et al. observed that breast cancer cells exposed to adriamycin in vitro can increase the release of L-EVs containing the transient receptor potential channel 5 (TrpC5), namely a receptor-activated non-selective calcium permeant cation channel. The up-regulation of TrpC5 in BC-derived L-EVs is responsible for adriamycin trapping, while the intercellular transfer of TrpC5 to recipient cells stimulates the production of the MDR1 protein, thus conferring chemoresistance to drug-sensitive cells [59].

Other putative mechanisms implicated in the drug resistance are the expression of surface receptors on the L-EVs functioning as decoys for monoclonal antibodies (moAbs). In this context, Simon et al. showed that L-EVs released by GBM cells in the presence of bevacizumab can reduce the amount of drug available by its trapping on the vesicle surface [64]. Such a mechanism of resistance mediated by L-EVs may be far more common since it has been reported also for anti-EGFR agents in a different context [40].

## 8. Applications of L-EVs for Liquid Biopsy

Liquid biopsy is currently under investigation as a diagnostic tool alternative to tissue biopsy in cancer. It consists of isolating tumor-derived components, such as circulating tumor cells (CTCs), circulating-free tumor DNA (ctDNA) and EVs, from blood or potentially any other biological fluid [4]. The downstream molecular analyses of these components may give a comprehensive overview of the tumor heterogeneity providing useful information for both prognostic and predictive purposes.

The use of EVs as liquid biopsy, particularly, is an fast growing field of research. Although L-EVs have been less investigated as compared to S-EVs, their peculiar molecular features and easiness of isolation render them a very attractive tool. In fact, while the ctDNA analysis refers only to small DNA sequences that are hard to discriminate from non-tumoral circulating-free DNA (cfDNA), the L-EV-DNA is more similar to the original cell DNA; thus, it is suitable for more complex analyses, such as epigenome-wide analysis, as well as methylation studies [17].

Moreover, circulating L-EVs are found in at least one order of magnitude higher as compared to the number CTCs detectable from single blood drawn [65] and therefore appear more suitable for downstream molecular analysis. The main uses of L-EVs as a liquid biopsy are described here and summarized in Table 2.

### 8.1. Diagnosis and Prognosis

A number of studies evaluated the possible use of L-EVs as cancer diagnostic biomarkers. Minciacchi et al., for example, proposed the measurement of CK-18 in L-EVs as a surrogate of the presence of PCa and found higher expression in plasma-derived vesicles from PCa patients when compared with healthy subjects [5]. However, these findings need further validation, since they were obtained in a small number of subjects, while the ubiquitous expression of CK-18 raises the question of the use of this protein as a PCa specific biomarker. Similar considerations must be made for other potential markers identified by the same authors to be enriched in tumor-derived L-EVs, that is, glyceraldehyde 3-phosphate dehydrogenase (GAPDH), glucose phosphate isomerase (GPI), lactate dehydrogenase B (LDHB) and heat shock protein 5 (HSPA5), which are almost ubiquitously expressed. On the other hand, by investigating the protein content of plasmatic L-EVs in a large and heterogeneous cohort of cancer patients, Menck et al. demonstrated the ubiquitously over-expression of the matrix metalloprotease inducer EMMPRIN, suggesting its possible use as a pan-cancer diagnostic biomarker [65]. Moreover, circulating L-EVs can be assayed for prognostic purposes since high plasma levels of L-EVs correlated with worse disease stages. For example, Nanou et al. showed that patients with castration-resistant PCa whose circulating tumor-derived L-EVs were found below a pre-specified cut-off (~102 × L-EVs), achieved a median overall survival (OS) which was significantly increased as compared to patients exceeding that limit (23.0 vs. 8.1 months) [66]. This research was also extended to the role of L-EVs in other metastatic malignancies, including colorectal, non-small cell lung cancer and BC, thus confirming the value of tumor-derived L-EV count for prognostication [67]. 

### 8.2. Predictive Biomarker

Since the mutational status of circulating EVs is considered an ideal surrogate of naive tumor cells, the molecular characterization of L-EVs isolated from body fluids may be an useful biomarker for precision therapy. In this context, limited evidence derives from a study with EVs purified from lymphatic drainage or exudative seroma, obtained after lymphadenectomy from 17 patients with stage III melanoma [68]. The Authors found that lymphatic drainage is a biofluid enriched in EVs, as compared to plasma and revealed a 50% concordance of the BRAF mutational status relative to matched tissue samples using an allele-specific quantitative PCR. Similar applications of liquid biopsy in melanoma have been recently explored by our group, including the analyses of T-cell derived exosomes and CTCs [69,70], for the identification of patients amenable to immunotherapy or targeted therapy. Moreover, by means of Sanger sequencing, we have recently revealed (Figure 2) the presence of the BRAFV600 mutation in L-EV-DNA obtained from commercial melanoma cells (data unpublished), thus confirming the potential use of L-EVs as predictive biomarkers in melanoma and potentially also in any other malignancies. A rapid translation of these acknowledgments into the clinical setting, however, is not yet feasible, given the low sensitivity of the method, while ddPCR or NGS are currently under investigation to overcome these limits. 

Both the protein and mRNA content of L-EVs may be also investigated for predictive aims. For example, the expression of the Breast Cancer Resistance Protein (BCRP) in L-EVs from patients undergoing neoadjuvant treatment for locally advanced BC proved to be correlated with primary resistance to anthracycline-based chemotherapy, indicating its possible use as a negative predictive biomarker [71].

### 8.3. Monitoring Response and Acquired Resistance to Treatments

Since the circulating L-EV count reflects disease status, as well as relative tumor burden, its longitudinal evaluation during treatment could be used for monitoring response. In this context, Julich-Haertel et al. identified a particular subset of AnnexinV+/EpCAM+/ASGPR1+ L-EVs which were highly specific for the identification of hepatic malignancies, whose circulating levels rapidly decreased after curative tumor resection, thus suggesting their possible use as biomarkers of minimal residual disease [72]. In the same fashion, Kassam et al., by exploring the relative early changes in the number of CTCs and L-EVs, elaborated an interesting model to predict the response to neoadjuvant chemo-radiotherapy in locally advanced rectal carcinoma [73].

Finally, analyzing the cargo of the L-EVs is another potential tool for monitoring tumor cell changes during anti-tumor therapy, to early detect the onset of acquired drug resistance. The identification of specific L-EVs, in fact, can be the distinctive of chemoresistant tumor cells as occur for TrpC5+ L-EVs that were demonstrated in metastatic BC patients following adriamycin-based chemotherapy, while being undetectable in chemotherapy naïve patients [59].

## 9. Conclusions and Future Perspectives

L-EVs are interesting vesicles, with potential clinical applications as they play an important role in the processes of cancerogenesis, metastasis and drug resistance. Their content in protein, DNA and regulatory RNAs, is several-fold higher than any other vesicle class, mirroring the composition of the parental cell. All these characteristics make L-EVs an ideal tool for liquid biopsy in oncology.

At present, promising results for the use of circulating L-EVs, as either a prognostic or predictive tool, have been produced with GBM and melanoma. Due to its intracranial localization, in fact, GBM is hard accessible for traditional biopsy and, thus, L-EVs liquid biopsy was proposed for prognostic stratification of patients, as well as to predict response to targeted therapy. On the other hand, the L-EV-DNA analysis has been proposed for the mutational analysis of melanoma patients amenable to anti-BRAF therapies. Moreover, mutational tracking along the therapy is thought to early detect the onset of acquired resistance with potential implications in the treatment decision.

Despite the original optimism for the translatability of L-EVs in the clinical arena, several limitations need to be first overcome. In this context, the major challenge concerns the lack of standardized protocols for L-EVs isolation and downstream analyses, while the current absence of universally accepted markers for quality control restricts their use in the pre-clinical setting. Additionally, another unsolved issue depends on the contamination of tumor-derived L-EVs with the EVs that have not a tumoral origin, meaning that tumoral products result greatly diluted in the multitude of “physiological” L-EVs and hardly accessible to further molecular testing. 

A possible solution could be the use of highly specific analytical techniques able to differentiate the tumor-derived material from the physiological material. In this direction, instruments for single EVs analysis are under development [74]. Moreover, the implementation of novel immune affinity tools may allow the selective isolation of tumoral L-EVs based on the recognition of specific cancer-derived surface markers, while the genome amplification and highly sensitive NGS techniques should be used for the molecular analyses of these vesicles.

In conclusion, although the pre-clinical data of L-EVs appear very promising, further methodological improvements and validation from large clinical trials are needed to support the applicability of these vesicles as tumor biomarkers for tailoring treatment decisions or monitoring tumor progression.

## Figures and Tables

**Figure 1 ijms-21-06543-f001:**
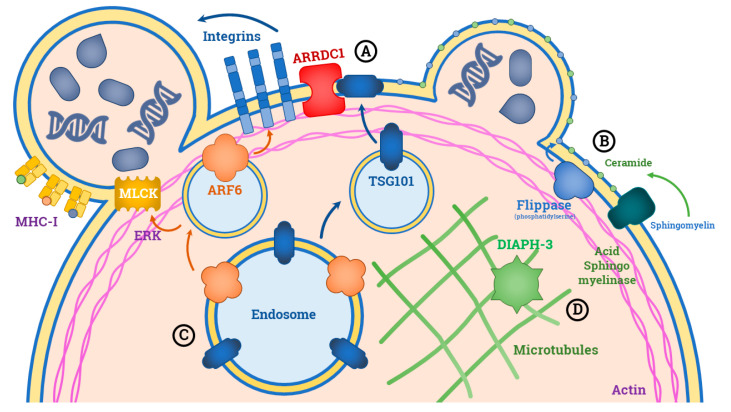
Principal mechanisms involved in the biogenesis of large extracellular vesicles (L-EVs). Many pathways are involved in L-EVs’ generation. (**A**) The Arrestin Domain-Containing Protein-1 (ARRDC1) induces the relocalization of TSG101 from the endosomal compartment to the plasma membrane, thus provoking shape changes in the cell membrane curvature that initiate the microvesicle gemmation. (**B**) Similar membrane plasticity modifications depend on the translocation of phosphatidylserine on the outer membrane layer or by the acid sphingomyelinase-mediated formation of ceramide. (**C**) The ADP-ribosylation factor 6 (ARF6) can influence the incorporation of integrins and the Major Histocompatibility Complexes type I (MHC-I) into the microvesicles. It also recruits the myosin light-chain kinase (MLCK) via ERK, thus initiating the outward budding of the plasma membrane. (**D**) Parallel, the generation of membrane blebs of 1–10 μm in diameter is mediated by the Diaphanous-related formin-3 (DIAPH3) inactivation, namely a cytoskeletal regulating protein often down-regulated in cancer.

**Figure 2 ijms-21-06543-f002:**
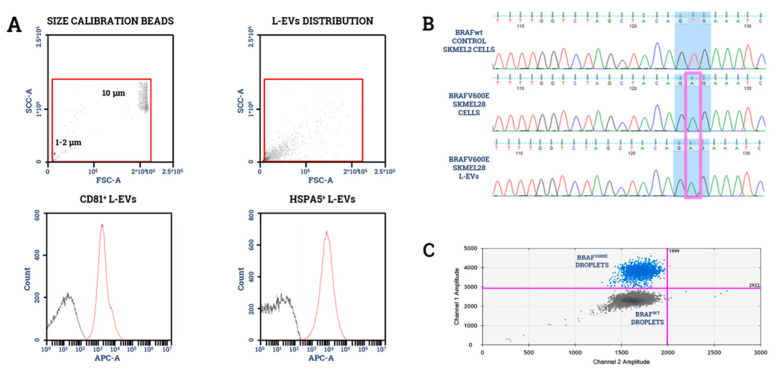
BRAF mutational analysis of L-EV-DNA. Representative BRAFV600E mutational analysis completed on L-EVs isolated by ultracentrifugation from commercial SK-MEL28 melanoma cells. (**A**) As shown, L-EVs were characterized by flow-cytometry for size in relation to their disposition by FW/SSC scatter as well as expression of HSPA5 and CD81. (**B**) Following the DNA extraction, we demonstrated by Sanger sequencing the presence of the BRAFV600E mutation in both SK-Mel28 parental cells and relative L-EVs, as compared to BRAF wild type control cells (SK-MEL2). The triplet referred to the V600 position is shaded in blue, while the mutated base is contained in the purple box. (**C**) Similar findings on SK-MEL28 derived L-EVs were also obtained using the droplet digital-PCR. Blue dots represent the BRAFV^600E^ droplets, while gray dots represent the BRAF^WT^ droplets.

**Table 1 ijms-21-06543-t001:** Methods for L-EVs isolation in comparison.

L-EV Isolation Method	Advantages	Disadvantages
**Ultracentrifugation**	Cost-effective No limitations on sample volume No additional reagents needed	Time-consuming Possible protein contamination
**Ultrafiltration**	Easy to perform No limitations on sample volume	EVs can be damaged by pressure Possible protein contamination
**Gel Filtration**	Easy to perform Preservation of EV integrity	High cost Diluted yield Possible protein contamination
**Precipitation**	Fast to perform No limitations on sample volume	Possible protein contamination
**Immunoaffinity**	Isolation of specific EVs subpopulations High purity of the yield	Possible nonspecific binding Lower recovery efficiency

**Table 2 ijms-21-06543-t002:** Possible application of L-EVs liquid biopsy.

Applications for L-EV Liquid Biopsy	Evidences
**Cancer diagnosis**	L-EVs containing CK18 as a marker of Prostate Cancer EMMPRIN+ L-EVs as a generic cancer marker
**Prognosis evaluation**	Higher plasma L-EV count correlates with poorer prognosis (PCa, BC, NSCLC, CRC)
**Predictive biomarker**	The analysis of predictive mutations could be performed on plasma L-EVs
**Treatment monitoring**	AnnexinV+/EpCAM+/ASGPR1+ L-EVs as a marker of response to surgery in HCC TrpC5+ L-EVs as a marker of chemoresistance in BC BCRP+ L-EVs as an early marker of chemoresistance in BC

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
