# Peer review of "Large Extracellular Vesicles—A New Frontier of Liquid Biopsy in Oncology"

_ijms, 2020, doi:10.3390/ijms21186543_

Round 1

Reviewer 1 Report

This review article is well written and interesting.

Several minor revisions are required.

1) Fig. 1 (illustration): MHC-II, ARDCC1, phosphatidylserin, Sphyngomyel, and Sphyngomyelinase should be MHC-I, ARRDC1, phosphatidylserine, Sphingomyelin, and Sphingomyelinase, respectively.

2) Table 1: Full spelling should be provided for Pros and Cons.

3) Line 427 (Abbreviations): All abbreviations used should be listed here.

Author Response

1) The Reviewer noticed some spelling errors in Fig 1.

Corrected as appropriate.

2) The Reviewer required the full spelling for “Pros” and “Cons” in Table 1.

The headings of Table 1 have been corrected.

3) The Reviewer noticed some abbreviations used along the text which were missing in the dedicated section.

The “Abbreviations” section (line 436) has now been completed.

Reviewer 2 Report

The present manuscript entitled “Large Extracellular Vesicles: a new frontier of liquid biopsy in oncology” covers many topics and issues concerning the use of extracellular vesicles as “liquid biopsy” tools. The manuscript is well written and the paragraphs are given with the appropriate order. However, the authors should give more information regarding the EVs derived from tumor microenvironment and their contribution to tumor progression and metastasis. Furthermore, since host immune response is a major issue for carcinogenesis, tumor progression and metastasis, the authors should provide more information regarding the role of EVs derived from the different subtypes of immune cells and their potential role in the periphery

Author Response

1) The Reviewer claimed more information regarding the EVs derived from tumor microenvironment and their contribution in tumor progression and metastasis.

As required, we have now implemented the text with more data about the role of tumor-microenvironment derived L-EVs in tumor progression (lines 282-289).

Appropriate references have been included (Richards KE, et al. Oncogene 2017; Hu Y, et al. PLoS One 2015; Ji R, et al. Cell Cycle 2015).

2) The Reviewer suggested to include additional information regarding the role of L-EVs from immune cells.

The contribution of leucocyte-derived L-EVs in cancer progression is clearly a topic of interest. Unfortunately, the data currently available are only relative to the S-EVs. Thus, we are thankful to the Reviewer for his observation and we added a new comment on this topic (lines 295-300), but we remarked the absence of evidence that limits definite conclusions.

Two new references have also been included (Nanou A, et al. Cells 2019; Oggero S, et al. Front Pharmacol 2019).